# Towards a Comprehensive Framework for Made-to-Measure Alginate Scaffolds for Tissue Engineering Using Numerical Simulation

**DOI:** 10.3390/gels11030185

**Published:** 2025-03-07

**Authors:** Alexander Bäumchen, Johnn Majd Balsters, Beate-Sophie Nenninger, Stefan Diebels, Heiko Zimmermann, Michael Roland, Michael M. Gepp

**Affiliations:** 1Applied Mechanics, Saarland University, Campus A4 2, 66123 Saarbruecken, Germany; alexander.baeumchen@uni-saarland.de (A.B.); s.diebels@mx.uni-saarland.de (S.D.); 2Fraunhofer Institute for Biomedical Engineering (IBMT), Joseph-von-Fraunhofer-Weg 1, 66280 Sulzbach, Germany; johnn-majd.balsters@ibmt.fraunhofer.de (J.M.B.); beate-sophie.nenninger@ibmt.fraunhofer.de (B.-S.N.); heiko.zimmermann@ibmt.fraunhofer.de (H.Z.); michael.gepp@ibmt.fraunhofer.de (M.M.G.); 3Molecular and Cellular Biotechnology/Nanotechnology, Saarland University, Campus Saarbruecken, 66123 Saarbruecken, Germany; 4Faculty of Marine Science, Universidad Católica del Norte, Larrondo 1281, Coquimbo 1780000, Chile

**Keywords:** alginate hydrogels, tissue engineering, scaffolds, cross-linking, multi-phase modeling, finite element method (FEM)

## Abstract

Alginate hydrogels are integral to many cell-based models in tissue engineering and regenerative medicine. As a natural biomaterial, the properties of alginates can vary and be widely adjusted through the gelation process, making them versatile additives or bulk materials for scaffolds, microcarriers or encapsulation matrices in tissue engineering and regenerative medicine. The requirements for alginates used in biomedical applications differ significantly from those for technical applications. Particularly, the generation of novel niches for stem cells requires reliable and predictable properties of the resulting hydrogel. Ultra-high viscosity (UHV) alginates possess alginates with special physicochemical properties, and thus far, numerical simulations for the gelation process are currently lacking but highly relevant for future designs of stem cell niches and cell-based models. In this article, the gelation of UHV alginates is studied using a microscopic approach for disc- and sphere-shaped hydrogels. Based on the collected data, a multiphase continuum model was implemented to describe the cross-linking process of UHV alginate polysaccharides. The model utilizes four coupled kinetic equations based on mixture theory, which are solved using finite element software. A good agreement between simulation results and experimental data was found, establishing a foundation for future refinements in the development of an interactive tool for cell biologists and material scientists.

## 1. Introduction

### 1.1. Alginate Hydrogels

Alginate polysaccharides derived from brown algae are often used as excipients or bulk materials in biotechnological and biomedical products. In biomedical applications, these anionic polymers serve as immobilization matrices for enzymes or cells, carriers for controlled drug release, wound dressings and dental impression materials.

In medical biotechnology, alginate hydrogels and their derivatives are used as scaffolds, microcarriers and immobilization matrices, generating defined cell niches for physiologic cell-based models [1], e.g., for disease modeling [2] or drug discovery [3]. Furthermore, alginate beads, modified with extracellular matrix proteins, are used in suspension bioreactors for expanding anchorage-dependent stem cells [4].

The broad industrial use of alginates is attributed to their unique properties: They are non-toxic, biocompatible and gel in mild conditions (ambient temperature, physiological pH, low concentrates and non-toxic cross-linking compounds).

Chemically, alginates are composed of unbranched polymers of β-D-mannuronate (M) and α-L-guluronate (G), linked via glycosidic bonds. The G/M ratio and the composition of homogeneous or heterogeneous polymer blocks vary by algae species and their respective environments. During cross-linking, mainly G-blocks interact with multivalent cations such as Ca^2+^, Ba^2+^ and Fe^2+^, forming the “egg-box” structure, resulting in hydrophilic hydrogels with water content over 98% [5]. The mechanical properties of gelled hydrogels depend on parameters such as the type of cross-linking ion [6], its concentration, or cross-linking time. In the past decades, ultra-high viscosity (UHV) alginates [7,8] have been extensively studied and established in regenerative medicine and tissue engineering [9]. In contrast to other commercially available alginates, UHV alginates have tenfold longer polymers while possessing the same chemistry and cross-linking mechanisms [9]. Furthermore, some UHV alginates are characterized by sterility, low endotoxin levels and high purity [10].

These properties result in low immunogenicity, high cell compatibility, high mechanical stability, higher chemical resistance and higher elastic modulus, making this biomaterial suitable for applications in medical biotechnology.

### 1.2. Simulation/Model of Alginate Hydrogel Formation

Over the past few decades, simulations have provided valuable insights into the formation of alginate hydrogels, often, for simplification, focusing on a few or just a single aspect of the process. Stewart et al. [11] created a molecular dynamics (MD) simulation to investigate the complexation of poly-M and poly-GM oligomers with sodium and calcium ions, forming an alginate gel. Previous studies focused primarily on poly-G sequences [12,13,14]. They observed that all conducted simulations resulted in complexation with Na^+^ and Ca^2+^ ions, regardless of the oligomer sequence used. The formation of the “egg-box” structure appeared to follow the order of GG > MM > GM in terms of reaction preference, suggesting that poly-M regions may play a role in secondary cross-linking interactions.

Nair et al. [15] developed an in silico model to determine the mechanical properties of alginate-based 3D scaffolds, identifying the Ogden hyperelastic model as the most suitable for describing the nonlinear behavior of alginates. This model, modified to include alginate concentration and viscosity, allowed analysis of physical and geometric parameters such as porosity on the stress-strain curve, showing good agreement with experimental data.

Külcü et al. [16] created a micro-mechanical constitutive model for the nonlinear inelastic properties of alginate double network hydrogels, employing the Arruda–Boyce eight-chain model [17], network decomposition [18,19] and network alteration concepts [20]. This model also demonstrated good agreement with experimental results.

Mikkelsen et al. [21] proposed a mathematical model for calcium-induced 3D alginate gel formation using coupled chemical reaction-diffusion equations. They used measurements from 1D gels to provide the necessary parameter values. Thu et al. [22] took experimentally determined gradients, gathered through MR microimaging and SRIXE, from 3D gels to reevaluate these previous parameter estimates, e.g., the relative alginate diffusion coefficient and the reaction rate constant. Bjørnøy et al. presented an experimental framework for the gelation of low viscosity alginate with calcium concentrations higher than 50 mM. Their main focus is to understand the gelation and mineralization of alginate hydrogels; only minor emphasis is placed on the numerical modeling of this process [23].

### 1.3. Analysis of Hydrogel Cross-Linking Process

So far, numerous in vitro studies have been conducted to understand the correlation between the type and concentration of the cross-linker and the resulting hydrogels. Since it is very challenging to differentiate between the transitions of sol and gel states of hydrogels, researchers have found several approaches to quantify this process. Huynh et al. studied the gelation kinetics of polygalacturonate with different divalent cations (Mg^2+^, Ca^2+^, Zn^2+^ and Ba^2+^) using turbidity changes during the sol–gel transition, determining the diffusion coefficients of different cross-linker concentrations [24]. However, this method’s reliance on turbidity limits its applicability across all hydrogels. Another approach for visual analysis of ion occurrence, primarily applicable to Ca^2+^-induced processes, was presented by Cheng et al. In the context of an electro-induced gelation mechanism, they utilized a pH-dependent release of Ca^2+^ ions from CaCO_3_ and monitored the gelation process with a Ca^2+^-dependent fluorescent dye [25,26]. Photon Correlation Imaging was employed to investigate the cross-linking process in small volumes of alginates by Secchi et al. [27] using Ca^2+^. Hajikhani et al. proposed a chemo-mechanical model for alginate cross-linking based on their experimental data with commercial sodium alginate and Ca^2+^ ions for bioprinting applications. They focused on diffusion and reaction kinetics, incorporating the resulting finite strains and shrinkage effects caused by syneresis and additionally accounting for the cross-linking reaction rate dependence on internal gelling mechanics, establishing a two-way coupling between chemistry and mechanics [28,29]. Stößlein et al. employed mechanical texture analysis to determine in situ cross-linking of alginates with Ca^2+^, correlating guluronate content with the applied pressure on compressed beads [30]. Cesewski et al. described in their study a cantilever-based method for real-time monitoring of the alginate cross-linking process using Ca^2+^ ions [31], and Potter et al. utilized magnetic resonance imaging to study Ca^2+^-based cross-linking [32]. Bjørnøy et al. analyzed the cross-linking kinetics of low-viscosity alginates in the presence of Ca^2+^, imaging the cross-linking front by dark-field and confocal laser-scanning microscopy of regularly formed alginate discs or beads [23]. They used a modified numerical model originally presented by Thu et al. [22]. Furthermore, they studied only low- and medium- viscosity alginates and used cross-linker concentrations higher than 50 mM [23]. Hu et al. investigated diffusion properties using fluorophore-labeled albumin in various alginate formulations, but did not perform numerical simulations [33]. Li et al. measured swelling, transparency and FT-IR spectra during alginate gelation, but only tested low- and medium-viscosity alginates without numerical modeling [34].

Besiri et al. used a rheometer approach with a modified rheometric setup to analyze the gelation of technical low-viscosity alginate using CaCl_2_ [35].

An assessment of the polymer gradients using synchrotron radiation-induced X-ray emission, magnetic resonance microimaging and mathematical modeling [22] was conducted by Thu et al. They primarily focused on the inhomogeneity of alginate microbeads and not on gelation kinetics. Furthermore, ultra-high-viscosity alginates were not considered in the study, and gelation was conducted with Ca^2+^-ions.

The numerical model presented in this study employs a reaction-diffusion approach to the gelation process. While others [23,28] have taken, modified and expanded on the framework developed by Mikkelsen and Elgsaeter [21], this work aims to explore a different numerical approach. Additionally, this study focuses on barium instead of calcium as the cross-linking ion in the process of egg-box formation.

### 1.4. Alginate Hydrogels in Tissue Engineering and Regenerative Medicine

In biotechnological applications, alginate hydrogels have been used for several decades primarily as an immobilization matrix for therapeutically relevant cells [36,37,38,39]. The discovery of induced pluripotency [40,41] revolutionized regenerative medicine by enabling the derivation of any human somatic cell type in vitro. This advancement heightened the demand for cell type-specific environments that more accurately replicate in vivo conditions compared to traditional plastic dishes [42,43,44].

The rational design of the stem cell niche to provide tissue-specific environments combines soluble factors, adjacent cell types, specific proteins of the extracellular matrix, the topography of the adhesion site and the mechanical properties of the bulk material. Cell-type-adapted niches lead to in vivo-like behavior of the cells and, in consequence, to valuable data when these cells are used as models for drug screening, cytotoxicity and disease modeling. Cultivating cells in environments different from their native conditions, such as stiff 2D surfaces instead of 3D environments [42,45], can lead to inaccurate dose responses in drug testing, as demonstrated in breast cancer research by Imamura et al. [46].

### 1.5. Rationale Predicting Alginate Hydrogel Gelation

In cell biology, protocols for generating soft 2D and 3D environments rely heavily on concentration and time, with final hydrogel properties often characterized empirically. Variations in gelation conditions usually lead to relative characterizations such as “softer” or “stiffer”. Besides the chemical conditions of the reaction, the volume of the alginate to be gelled can be altered, e.g., if the cell culture substrate must be changed. Higher volumes with the same gelling conditions lead to inhomogeneous or only partially gelled hydrogels. In general, the produced hydrogels must be reliable and their mechanical properties comparable, but intensive quality control and characterization are often not feasible, require special hardware and expertise and are, in addition, time-consuming.

The gelation history significantly affects the properties of cross-linked alginate constructs, as the diffusion properties of cross-linkers and the availability of alginate cross-linking sites change after cross-linking, leading to cross-linking irregularities and mechanical or chemical gradients in the gel [47,48,49].

Gelation is further influenced by encapsulated cells, whether single cells or multicellular aggregates, which can cause critical structures or inhomogeneous gelation, potentially triggering biological responses, such as in immuno-transplantation [10].

Overall, a predictive tool for the gelation of hydrogels is in high demand and would also be useful for cell biologists producing tissue-specific hydrogel surfaces or for quality control in regenerative medicine. A numerical model that can predict the gelation behavior and the mechanical properties of alginate hydrogels is still lacking. In this work, the fundamental numerical simulation for a comprehensive framework for made-to-measure alginate scaffolds will be presented. The fundamental gelation kinetics will be modeled based on experimental data and validated in initial real-world scenarios.

## 2. Results and Discussion

### 2.1. Alginate Gelation

The gelation process from the alginate sol to the alginate hydrogel was analyzed by tracking a thin traveling front line from the outer edge to the center of the alginate disc. In a non-invasive analytical approach, this process was quantified using phase contrast microscopy and subsequent manual image analysis. Representative time-lapse sequences of cross-linking experiments are illustrated in Figure 1a (10 mM BaCl_2_), Figure 1b (20 mM BaCl_2_) and Figure 1c (40 mM BaCl_2_). Due to the low contrast of the gelation interface, images in these figures were enhanced for better visibility by FFT Bandpass Filter of ImageJ 1.53h (Fiji, parameters “filter large structures down to” 50 pixels, filter small structures up to 3 pixels, tolerance of direction 5%, with activated autoscaling and saturate image option).

The data reveal that the general course of the traveling liquid/gelled interface is comparable for all studied cross-linker concentrations (10 mM, 20 mM and 40 mM; Figure 2a). The kinetics of all studied concentrations can be classified into two phases: The first phase at the beginning features a fast-traveling visible front line, followed by a phase with a slower but also linear course. In addition, we observed a more decreasing slope/velocity in Figure 2a with increased cross-linker concentration, resulting in a faster gelation time of the alginate disc (Figure 2b). The velocity of gelation in this work is defined as the reduction of the ungelled core and is negative.

The investigation of the cross-linking kinetics of UHV alginate spheres using three different BaCl_2_ concentrations with comparable osmolarities was performed by dropping 5 µL into the cross-linking bath. The cross-linking front was observed microscopically, and the advance of the front was measured by analyzing the diameter of the front over time. Since the sizes of the spheres differ from 1210 to 1705 µm, the initial sizes of the cross-linking fronts differ as well and require accordingly more or less time to reach the core of the spheres. The cross-linking fronts in the replicates with 10 mM BaCl_2_ (Figure 3a) need 145 to 290 s to reach the core and are strongly dependent on the sphere’s diameter. Figure 3b shows the diameter of the cross-linking front over time for a 20 mM BaCl_2_ cross-linking concentration. In all five replicates, the decrease in size over time was observed, with a steeper decrease than in Figure 3a. Since the sizes of the spheres differ from 1124 to 1885 µm in size the initial size of the cross-linking front differs as well and needs accordingly more or less time to reach the core of the spheres. The cross-linking fronts in replicates with 20 mM BaCl_2_ require 100 to 200 s to reach the core and are strongly dependent on the sphere’s diameter.

Figure 3c shows the diameter of the cross-linking front over time for a 40 mM BaCl_2_ cross-linking concentration. In all five replicates, the decrease in size over time was observed, with the steepest decrease compared to Figure 3a,b. The sizes of the spheres have comparable diameters (1790 to 1940 µm). Accordingly, the initial sizes of the cross-linking fronts are comparable as well and require almost the same time to reach the core of the spheres. The cross-linking fronts in the replicates with 40 mM BaCl_2_ need 120 to 150 s to reach the core and are also strongly dependent on the sphere’s diameter.

Overall, the tendency of the cross-linking velocities of discs and spheres is comparable (Figure 2b and Figure 3d). The velocity of gelation in this work is defined as the reduction of the ungelled core and is negative. The gelation results for spheres show the fastest cross-linking front for 40 mM BaCl_2_ with a subsequent decrease with decreasing BaCl_2_ concentration (40 mM > 20 mM > 10 mM, Figure 3d).

In this study, the gelation kinetics were investigated using an image-based approach and a numerical model created based on the chemical reaction and the collected data. The experimental data reveal a correlation between the cross-linker concentration, the velocity of the traveling gelation front and the overall gelation time. For the first time, the gelation kinetics of UHV alginates were analyzed in detail using the described optical approach. In the literature, the general phenomenon is described for low-viscosity alginates [23], but no systematic study of UHV alginates with a barium concentration lower than 50 mM had been conducted so far. In this study, the shape of the alginate was disc-like, as this allowed for higher throughput and better analysis. Although our preliminary data suggest that this type of analysis is also feasible for sphere-like, free-floating alginates, it requires a higher effort for analysis. The gelation kinetics reveal a strong dependency on the cross-linker concentration used. The main driver of this observation may be the diffusion of barium ions through the alginate network. The greater the initial cross-linking concentration, the higher the gradient from the outside to the interior of the hydrogel. Due to the high concentration, barium is available in excess for the gelation reaction, and in consequence, no slowdown of the gelation front occurred in the linear phase. The approach described in this study is furthermore capable of analyzing the influence of small cargo (cells or organoids) or debris (protein agglomerates) on gelation behavior.

The data also suggest shrinkage of the alginate discs, probably caused by syneresis (the release of water during gelation). This shrinkage is not yet accounted for in the current numerical model. Shrinkage (or syneresis, respectively) provides additional insights into the gelation kinetics, especially when two different alginate species are used. Alginates from LN contain more flexible heterogeneous MG/GM blocks, which enable coiling of polymers during gelation, leading to shrinkage of the hydrogel. Syneresis in alginates is complex, and the influence of the different components of gelation must be taken into account [50]. In further studies, the alginate composition, concentration of the cross-linking agent and shrinkage will be studied in detail to further enhance the current numerical model.

The course of alginate gelation in spheres is strongly dependent on barium concentration. This observation was expected, since a higher barium-concentrated solution, more ions are available to cross-link the alginate, and the time in which an ion finds a spot to cross-link a G-monomer is shorter. Therefore, it diffuses the fastest to the core of the UHV alginate sphere. At the lowest concentrated BaCl_2_ solution, the availability of barium ions is lower, and it takes more time to find a cross-linking spot; therefore, it takes more time for the cross-linker to reach the core of the UHV alginate sphere.

Overall, it seems that the optical observation of gelation kinetics is independent of the shape and volume of the alginates. Therefore, it will serve as a powerful and user-friendly experimental tool to further enhance the numerical model.

The microscopic images reveal a layer-like structure of the alginate discs. Over time, the number of layers at the outer surface increases. Similar structures have been described by Ehrhart et al. [51] and Jeong et al. [52]. In both studies, the depot of multivalent cations was the gelled alginate bead itself, which was suspended in liquid alginate. Thin alginate layers formed during gelation due to the diffusion of multivalent cations from the bead to the bead-alginate interface. A shell-wise gelation of alginate beads was also described in a review by Leong et al. [53] and Voo et al. [54]. In their study, Voo et al. found that highly concentrated (10% w/v) low-viscosity (LV) alginates form a multi-layer internal structure. Interestingly, this structure differs from low-concentration (2% w/v) LV-alginates. Low-concentration LV alginates resemble a more sponge-like structure with small cavities (pores). The different structures and varying degrees of homogeneity can be explained by the relative diffusion rates of alginate and the multivalent cation [54].

The experiments in this study were not designed to reveal the mechanism of layer formation during the gelation of UHV alginates. However, initial interesting findings were made, indicating that such layer formation can be observed non-invasively during gelation with phase-contrast microscopy (Figure 4a). In the given example, the thickness of the adjacent layers was approximately 15 to 30 µm under the 10 mM condition. These initial findings are summarized in Figure 4b. It can be hypothesized that the layer formation of UHV alginates differs from the findings of LV-alginates, as polymers with higher molecular weights lead to differences in layer thickness.

For a detailed analysis and correlation with cross-linker concentration, alginate composition and alginate concentration, a microscopic analysis with higher resolution and magnification must be conducted in future studies. By using alginates with different interactions with multivalent cations, these layers could help to understand the diffusion rates of the cations and polymers in UHV alginates. In this context, future models will also consider the step-wise interaction of barium ions with alginate chains, as presented for calcium by Zhao et al. in a population balance model [55]. This model will be integrated and combined with the findings of layer-wise gelation to more precisely project the properties of the alginate gels at the hydrogel’s surface.

### 2.2. Numerical Solution

A time-lapse sequence of the obtained simulation results for each cross-linker concentration (10 mM, 20 mM and 40 mM) is depicted in Figure 5, analogous to Figure 1. As previously mentioned, there is currently no shrinking behavior implemented in the model, leaving the alginate discs at the same diameter throughout the gelation process. These simulations are completed in similar amounts of time as the experimental tests, with 10 mM being slightly slower (3-min difference) and 20 mM and 40 mM being slightly faster (3-min and 4-min differences, respectively). This difference, however, is mostly observed at the end of the simulation, when the gelation front shows an increase in speed, which does not occur in the experiments (see Figure 6). Otherwise, the two phases of fast front propagation followed by a slower linear phase are also present here.

The simulation results of gelation in Figure 5 are also available in the Appendix A.

Overall, the presented model achieves a satisfactory level of agreement with the experimental data and offers a strong starting point for subsequent improvement and expansion.

The sudden acceleration toward the end of Figure 6 could be due to variations in mesh refinement between the center and the outer rim of the circular domain. These adjustments may also help resolve the formation of steps observed in the simulation results at smaller cross-linker concentrations. However, both issues are still being actively investigated. The spikes or abrupt increases at the very end of the graphs result from the automated front calculations detecting minor movements and simulation artifacts, as the primary reaction front has nearly dissipated.

Computation time can vary significantly depending on model parameters and the size, complexity and refinement of the used geometry mesh.

## 3. Conclusions

The experiments demonstrated the gelation of hydrogels devoid of cells, cargo or air bubbles, while also revealing the behavior of the cross-linking front upon encountering inclusions (debris). Images captured post-gelation showed swirling structures around these inclusions, which are critical to investigate for mechanical stability in high-density cell immobilization applications like bioprinting. The numerical model, based on gelation observations and stoichiometry, provides a foundation to optimize gelation properties. Future work will refine the model with nanoindentation data to predict the hydrogels’ Young’s modulus, potentially leading to a predictive tool (e.g., a mobile app) for tissue generation, aiding cell biologists and materials scientists in tissue engineering applications.

The results of this work will continually develop into a complex framework that will provide biologists and materials scientists with a virtual supervisor to produce highly defined alginate hydrogels for their applications. One of the next steps will be the integration of the mechanical properties of the material due to their immanent importance in stem cell biology and tissue engineering. The mechanical properties of native tissue vary from soft (e.g., brain) to ultra-stiff (e.g., bone) [56]. In alginates, gelation through multivalent cation interactions with guluronic acid blocks dictates their mechanical properties. By adjusting the chemical composition and reaction conditions, alginate hydrogels can be tailored to be softer or stiffer. These hydrogels can serve as soft growth surfaces for stem cells [1,4] or as immobilization matrices for therapeutically relevant cells [9]. Mechanical stimuli from the hydrogel environment, transmitted through cell-matrix contacts, can activate mechanotransduction pathways, thereby influencing cell behavior. For pluripotent stem cells, the rigidity-dependent YAP/TAZ signaling pathway plays a role in neural differentiation [57,58,59].

## 4. Materials and Methods

### 4.1. Disc-Shaped Alginate Gelation Kinetics Study

Alginate solutions from brown algae, *Lessonia nigrescens* and *Lessonia trabeculata,* were purchased from alginatec GmbH (Riedenheim, Germany). In this study, a 1:1 mixture (vol%/vol%) of 0.65% solutions (wt%/vol%) of both alginate species was used. The alginates were dissolved in saline (0.9%, B. Braun Melsungen AG, Melsungen, Germany). The dynamic viscosity of the mixture was characterized using the rheometer MCR 92 (Anton Paar GmbH, Ostfildern, Germany) and was 3306 (±78 mPa∙s). The different cross-linking solutions were prepared with 115 mM NaCl and 10, 20 and 40 mM BaCl_2_ (Merck KGaA, Darmstadt, Germany). The osmolarity of all cross-linking solutions was adjusted to 270 mOsm/L. Alginate solutions were pipetted onto Nunc EasYDish dishes (ThermoFisher Scientific GmbH, Dreieich, Germany) and covered with silicone spacers (Grace Bio Labs, Merck KGaA, Darmstadt, Germany, CoverWell incubation chamber, 20 μL volume, 13 mm diameter and 200 μm height). To facilitate the injection of cross-linking solutions, two thin channels were punched into the silicone wall (Figure 7a). Dishes and silicone spacers were cleaned with desalted water in an ultrasonic bath and air-dried before usage. One droplet of alginate (4 μL) was carefully pipetted into the middle of the petri dish and covered with a silicone spacer. Enclosed air bubbles occurred rarely and did not influence the gelation process (Figure 7b). Image acquisition was started, and 20 μL of the cross-linker solution was pipetted through the inlet to fill the whole chamber and start the gelation process. Time-lapse movies were acquired using a standard phase-contrast microscope (Eclipse Ts2, Nikon Instruments, Dusseldorf, Germany) at 4× magnification. Images were captured automatically with a mounted camera (DMK 33UX174 and software IC capture 2.4, both The Imaging Source Europe GmbH, Bremen, Germany) in the following intervals: 0 to 5 min: every 10 s, 5 min to 15 min: every 20 s and for the rest: every 40 s. Captured images were analyzed using the IC Measure software 2.0.0.161 (The Imaging Source Europe GmbH, Bremen, Germany).

### 4.2. Sphere-Shaped Alginate Gelation

Alginate solutions from the brown algae *Lessonia nigrescens* and *Lessonia trabeculata* were purchased from alginatec GmbH (Riedenheim, Germany). For the disc-shaped alginates, a 1:1 mixture (vol%/vol%) of 0.65% solutions (wt%/vol%) of both alginate species was used. The alginates were dissolved in saline (0.9%, B. Braun Melsungen AG, Melsungen, Germany). The dynamic viscosity of the mixture was characterized using a rheometer (MCR 102, Anton Paar GmbH, Ostfildern, Germany) and was 2041 (±12) mPa∙s.

To determine the cross-linking kinetics of alginate beads, different cross-linking solutions of barium chloride (Merck KGaA, Darmstadt, Germany) were prepared at concentrations of 10 mM, 20 mM and 40 mM. The osmolarity ranged from 291 to 301 mOsmol/L. Then, 100 µL of each cross-linking solution was transferred into a 96 U-bottom well plate (Greiner Bio-One GmbH, Frickenhausen, Germany). The addition of alginate to the cross-linking solution was performed by dropping 5 µL with a pipette. Video sequences of the cross-linking process were captured using a 4× phase contrast objective. For the observation of shrinkage, additional images were acquired after 180 min while the samples remained in their respective barium chloride solutions. A Nikon Eclipse microscope TS2 coupled with a DS-Ri2 camera (Nikon Instruments, Duesseldorf, Germany) was used for this analysis. The samples were kept at room temperature. Video sequences and images were processed using NIS-Elements Advanced Research (NIS AR 5.41.00, Nikon Instruments, Duesseldorf, Germany).

The cross-linking progress was analyzed every five seconds, starting from 15 to 20 s after dropping the alginate into the cross-linking bath. The cross-linking progress was observed as long as the cross-linking front was visible, for a maximum of 290 s. The diameters of the cross-linking fronts were plotted for all five replicates over time, and the speed of the cross-linking front was determined by linear fit.

### 4.3. Modeling

#### 4.3.1. Basic Assumptions

In the following investigation, it is assumed that the alginate hydrogel consists of a polymer $\varphi^{poly}$, an aqueous solvent $\varphi^{w}$ and barium ions $\varphi^{ba}$. During the cross-linking, the barium ions form bonds between the molecules of the polymer, resulting in the cross-linked polymer $\varphi^{cross}$ (see also Figure 8a). The model is developed on the basis of a multi-phase continuum description, i.e., a mixture theory is chosen as the starting point. Therefore, it is not the individual molecules or ions that are investigated but certain amounts of them occupying a representative volume element.

The amount of mass of the individual molecules is represented by the partial densities $\rho^{\alpha }$, where α ∈ {poly, w, ba, cross} indicates the above-mentioned constituents.

Figure 8b shows a representative volume of size (RVE) dv, which contains $N^{poly}$ mols of polymer molecules each of molar mass $M^{poly}$, $N^{ba}$ free barium ions of molar mass $M^{ba}$ and $N^{w}$ water molecules of molar weight $M^{w}$. Only one cross-linked polymer molecule is present in the RVE. Its molar mass increases during the cross-linking according to the number of polymers being added to the cross-linked one and the required number of barium ions. In consequence, the number of polymers and barium ions decreases during cross-linking. The water content is not influenced by the cross-linking because it acts as a solvent only.

The partial density of the constituents inside the RVE is defined as the ratio of their mass $dm^{\alpha }$ to the occupied volume dv. Using the number of molecules per volume $N^{\alpha }$ and the molar weight $M^{\alpha }$, the partial density of the constituent $\varphi^{\alpha }$ in the RVE is expressed as(1)$$\rho^{\alpha }=\frac{dm^{\alpha }}{dv}=\frac{N^{\alpha }M^{\alpha }}{dv}$$

While the total mass in the RVE is equal to the sum of the partial masses, the mixture density is obtained as(2)$$\rho =\sum_{\alpha }\rho^{\alpha }.$$

#### 4.3.2. Mixture Theory

In the following, a continuum mixture theory is applied as the basis of the model. The concept assumes that the RVE introduced in Section 4.3.1 is small compared to the macroscopic dimensions and, therefore, it is treated as a spatial point in the continuum description located at position x. This leads to the concept of superimposed continua because the RVE is simultaneously filled by all constituents, each of them described by its partial mass density.

In addition to the mass exchange between the constituents $\varphi^{poly}$, $\varphi^{ba}$ and $\varphi^{cross}$, the mass in the RVE can also change due to the flux of molecules/ions of the individual constituents over the boundary of the RVE. Balancing the mass flux $\rho^{\alpha }v^{\alpha }$ over the boundary of the RVE with the change of mass $\partial \rho^{\alpha }/\partial t\;$ inside the volume and the mass exchange ${\hat{\rho }}^{\alpha }$ leads to the local formulation of the mass balance of the constituent $\varphi^{\alpha }$:(3)$$\frac{\partial \rho^{\alpha }}{\partial t}+div(\rho^{\alpha }v^{\alpha })={\hat{\rho }}^{\alpha }.$$Here, $v^{\alpha }$ is the macroscopic velocity of the constituent $\varphi^{\alpha }$ and the operator *div* is the divergence with respect to the spatial position x. The situation is captured in Figure 8b, where a magnification of the point at location x shows the RVE.

Following Truesdell’s metaphysical principles of mixture theory, the balance of total mass is obtained by summing up the mass balances of the individual constituents. Furthermore, the format of this balance must be identical to the balance equation of a single-phase continuum. Summation of (3) yields(4)$$\frac{\partial \sum_{\alpha }\rho^{\alpha }}{\partial t}+div\sum_{\alpha }\rho^{\alpha }v^{\alpha }=\sum_{\alpha }{\hat{\rho }}^{\alpha }.$$

This result is identical to the mass balance of a single-phase continuum if (2) is substituted into the first term and if the barycentric velocity or mixture velocity is defined according to(5)$$\rho v=\sum_{\alpha }\rho^{\alpha }v^{\alpha }.$$

Furthermore, the total mass is conserved. This requires that the sum of the mass exchange terms to be zero:(6)$$\sum_{\alpha }{\hat{\rho }}^{\alpha }=0.$$

In the same manner, further balance equations can be derived, e.g., the balances of momentum and energy, cf. e.g., Truesdell [60], Bowen [61], etc.

As a simplification, it is assumed that during the cross-linking process, the barycentric velocity is zero, i.e., there is no average macroscopic motion. However, the individual constituents may move. To describe this relative motion of $\varphi^{\alpha }$ with respect to the center of mass of the mixture, which remains at rest, the diffusion velocities are introduced by(7)$$d^{\alpha }=v^{\alpha }-v.$$

Using the definition (2) of the mixture mass density and the definition (5) of the barycentric velocity, it can be proven that(8)$$\sum_{\alpha }\rho^{\alpha }d^{\alpha }=0.$$

With the definition of the barycentric velocity (5) and the diffusion velocity (7), the mass balances can be rewritten as(9)$$\frac{\partial \rho^{\alpha }}{\partial t}+div\left(\rho^{\alpha }v+\rho^{\alpha }d^{\alpha }\right)={\hat{\rho }}^{\alpha }.$$

In further investigations, it is assumed that the diffusion velocities and their changes are small. Therefore, the associated change in momentum is small. This avoids a detailed study of the balances of momentum for the individual constituents and the diffusive mass fluxes.(10)$$j^{\alpha }:=\rho^{\alpha }d^{\alpha }$$
can be prescribed by constitutive equations. In the present case, diffusion is the main driving force. Therefore, the flux terms are governed by Fick’s second law. For simplicity, the barycentric motion is not further investigated during the cross-linking, i.e., no external loads are applied to the mixture,(11)v=0.

Based on the above-mentioned assumptions, the cross-linking model consists of the four balance-of-mass equations(12)$$\frac{\partial \rho^{poly}}{\partial t}=-divj^{poly}+{\hat{\rho }}^{poly},$$(13)$$\frac{\partial \rho^{w}}{\partial t}=-divj^{w}+{\hat{\rho }}^{w},$$(14)$$\frac{\partial \rho^{ba}}{\partial t}=-divj^{ba}+{\hat{\rho }}^{ba},$$(15)$$\frac{\partial \rho^{cross}}{\partial t}=-divj^{cross}+{\hat{\rho }}^{cross}$$
and the constraints(16)$$\sum_{\alpha }{\hat{\rho }}^{\alpha }=0,$$(17)$$\sum_{\alpha }\rho^{\alpha }d^{\alpha }=0.$$

#### 4.3.3. Constitutive Equations

For the following investigations, the aqueous suspension is neglected, and only the polymer, cross-linked polymer and barium constituents are investigated. During the cross-linking, the mass density of the cross-linked polymer increases, while the densities of the barium ions and the polymers decrease. Considering the constraint (6), the mass production of the cross-linked polymer is given by(18)$${\hat{\rho }}^{cross}=-{\hat{\rho }}^{poly}-{\hat{\rho }}^{ba}.$$

The probability of the formation of a new node in the cross-linked polymer network increases with the amount of available barium ions and with the amount of free polymers. These effects are reflected in the constitutive assumptions(19)$${\hat{\rho }}^{poly}=-C^{poly}\rho^{poly}\rho^{ba},$$(20)$${\hat{\rho }}^{ba}=-C^{poly}\rho^{poly}\rho^{ba}-C^{cross}\left(1-\kappa \right)\rho^{cross}\rho^{ba}.$$

While one polymer can link several ions to the network, a second term is introduced in (20) with the following interpretation: While $C^{poly}$ describes the initial connection of one polymer to the cross-linked polymer, which requires one barium ion, a further cross-linking can take place. After the polymer becomes part of the cross-linked polymer, the remaining free positions can further absorb ions until all possible positions are occupied by barium ions and the maximum number of cross-links is formed in the cross-linked polymer. This effect is described by the additional internal variable κ, defined as the ratio of occupied bindings to possible bindings. The term $\left(1-\kappa \right)\rho^{cross}$ is a measure of the free binding positions. The evolution of κ is driven by a reaction equation:(21)$$\frac{\partial \kappa }{\partial t}=C^{\kappa }\left(1-\kappa \right)\rho^{cross}\rho^{ba}.$$

During the formation of the polymer network, the diffusion motion of the cross-linked polymer is neglected, i.e.,(22)$$d^{cross}=0.$$

This assumption implies that swelling of the cross-linked polymer is currently not included in the model for simplicity. In contrast, the polymer constituent and the barium ions move with respect to the stationary cross-linked polymer. When the motion is mainly driven by diffusion, an appropriate choice for the mass fluxes is Fick’s law:(23)$$\rho^{poly}d^{poly}=-D^{poly}grad\rho^{poly},$$(24)$$\rho^{ba}d^{ba}=-D^{ba}grad\rho^{ba}.$$

The size of ions is much smaller than the size of the polymers; therefore, the diffusion constant $D^{ba}$ is several orders of magnitude larger than $D^{poly}$:(25)$$D^{ba}\gg D^{poly}.$$

During the cross-linking, the mesh size of the cross-linked polymer decreases. This drastically hinders the diffusion motion of the polymers as well as the diffusion motion of the barium ions. For both diffusing constituents, an exponential function is chosen, relating the current value $D^{\alpha }$ of the diffusion constant to its initial value $D_{0}^{\alpha }$:(26)$$D^{\alpha }=D_{0}^{\alpha }\exp\left(-K^{\alpha }\rho^{cross}\right)\;for\;\alpha \in \left\{poly,ba\right\}.$$

Due to the different sizes of the ions and the polymers, the constants $K^{\alpha }$ are different.

### 4.4. Implementation

A set of kinetic differential equations based on the previous section is used to describe the local accumulation and depletion of mobile alginate, cross-linked alginate and cross-linking cations [62]. These kinetic differential equations are given as a set of diffusion-reaction Equations (27)–(30):(27)$$\frac{\partial }{\partial t}\rho^{poly}\left(t,x\right)=div\left(D_{0}^{poly}\exp\left(-K^{poly}\rho^{cross}\left(t,x\right)\right)div\rho^{poly}\left(t,x\right)\right)\;-C^{poly}\rho^{poly}\left(t,x\right)\rho^{ba}\left(t,x\right)$$(28)$$\frac{\partial }{\partial t}\rho^{ba}\left(t,x\right)=div\left(D_{0}^{ba}exp\left(-K^{ba}\rho^{cross}\left(t,x\right)\right)div\rho^{ba}\left(t,x\right)\right)\;-C^{ba}\rho^{poly}\left(t,x\right)\rho^{ba}\left(t,x\right)\;-C_{\kappa }^{ba}\rho^{ba}\left(t,x\right)\left(1-\kappa \left(t,x\right)\right)\rho^{cross}\left(t,x\right)$$(29)$$\frac{\partial }{\partial t}\rho^{cross}\left(t,x\right)=\left(C^{poly}+C^{ba}\right)\rho^{poly}\left(t,x\right)\rho^{ba}\left(t,x\right)\;+C_{\kappa }^{ba}\rho^{ba}\left(t,x\right)\left(1-\kappa \left(t,x\right)\right)\rho^{cross}\left(t,x\right)$$(30)$$\frac{\partial }{\partial t}\kappa \left(t,x\right)=C^{\kappa }\rho^{ba}\left(t,x\right)\left(1-\kappa \left(t,x\right)\right)\rho^{cross}\left(t,x\right)$$

This numerical model was implemented using the FEM software COMSOL Multiphysics^®^ v6.2 (COMSOL AB, Stockholm, Sweden). The simulations are run on a workstation containing a 16-core 11th Gen Intel^®^ Core™ i9-11900K CPU and 64 GB of RAM.

The model parameters, which are given in Table 1, were optimized to achieve the best fit between the simulation results and the experimental data.

## Figures and Tables

**Figure 1 gels-11-00185-f001:**
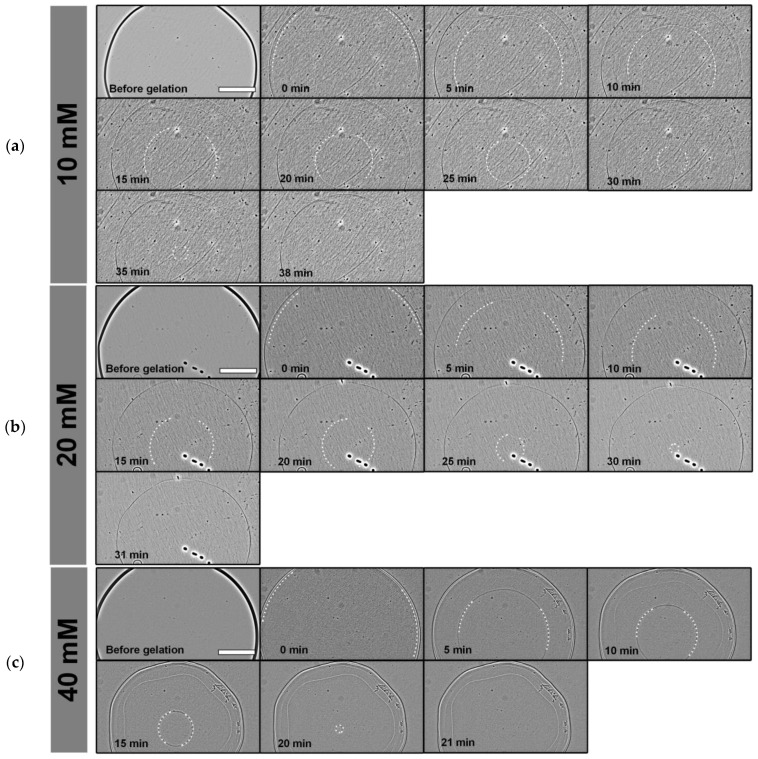
Time-lapse sequence of alginate gelation with different concentrations of cross-linking agents. (**a**) 10 mM BaCl_2_ solution, (**b**) 20 mM BaCl_2_ solution and (**c**) 40 mM BaCl_2_ solution. The gelation kinetics of the alginate are derived from the course of the traveling gelled/liquid interface. Due to low contrast, dashed white lines are used to indicate segments of the gelled/liquid interface. Scale bar indicates 1000 μm. Images are enhanced using a bandpass filter in ImageJ.

**Figure 2 gels-11-00185-f002:**
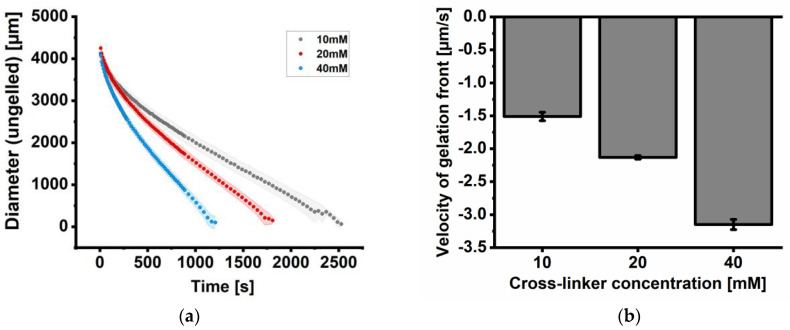
Analysis of the gelation process of alginate discs. (**a**) Gelation kinetics analyzed by the decreasing diameter of the gelation front. The kinetics of gelation depend strongly on the applied cross-linker concentration: the higher the BaCl_2_ concentration, the faster the overall gelation of the alginate droplet. (**b**) Velocity of the gelation front of alginates. Doubling the cross-linker concentration leads to a linear increase in gelation velocity. The velocity of gelation in this work is defined as the reduction of the ungelled core and is negative. Data are expressed as mean value ± standard deviation (n = 5 gelation experiments). Standard deviation in (**a**) is shown as a ribbon for visualization purposes.

**Figure 3 gels-11-00185-f003:**
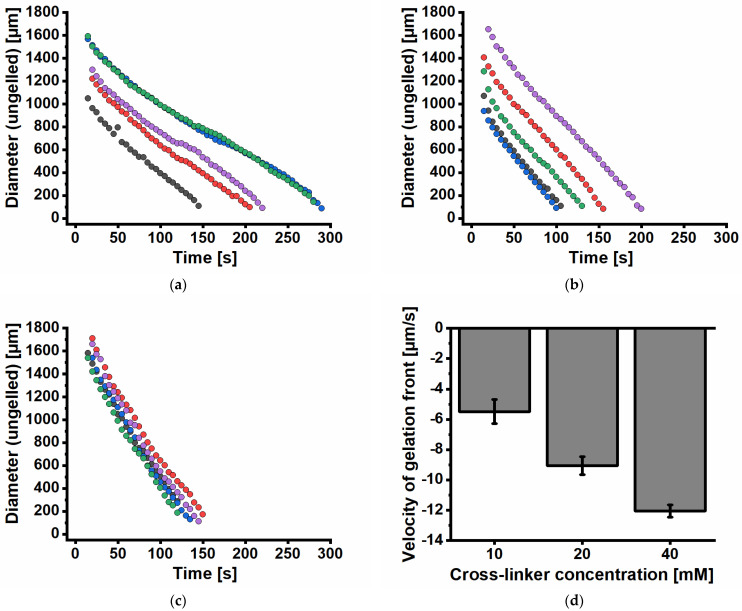
Analysis of the alginate gelation process of alginate spheres (beads, microcarriers). Gelation kinetics were analyzed by the decreasing diameter of the gelation front. The kinetics of gelation depend strongly on the applied cross-linker concentration: the higher the BaCl_2_ concentration, the faster the overall gelation of the alginate droplet. (**a**) Single gelation experiments using 10 mM BaCl_2_ solution; (**b**) single gelation experiments using 20 mM BaCl_2_ solution; (**c**) single gelation experiments using 40 mM BaCl_2_ solution; (**d**) the velocity of gelation front of alginates from (**a**) to (**c**) extracted by linear curve fitting. The velocity of gelation in this work is defined as the reduction of the ungelled core and is negative. Doubling the cross-linker concentration leads to a linear increase in gelation velocity. Data colors in (**a**–**c**) refer to different gelation experiments. Data in (**d**) are expressed as mean values ± standard deviation (n = 5 gelation experiments).

**Figure 4 gels-11-00185-f004:**
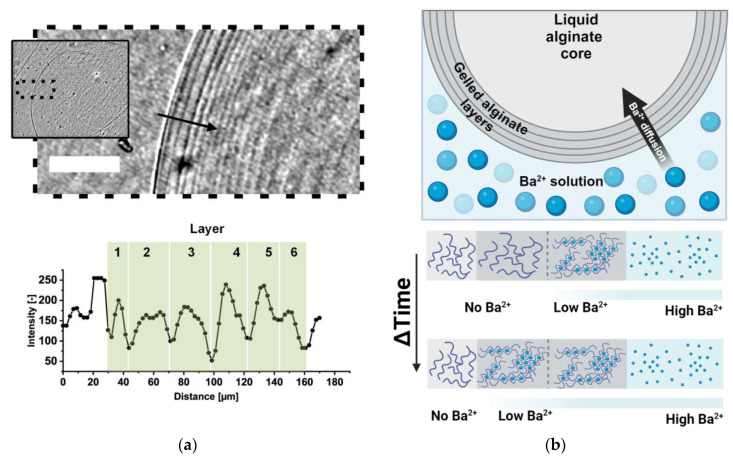
Alginate micro-layer formation during gelation. (**a**) **Top**: Microscopic image of the formed layer at the outer border of the alginate disc; scale bar: 200 µm. Inset: Lower magnification of the area indicated by the black dashed line. Black arrow: Line scan of intensity in the graph. **Bottom**: The graph illustrates the data from the line scan of intensity. (**b**) Schematic illustration of layer formation in alginate disc-like hydrogels (adapted from [52]; created with BioRender.com).

**Figure 5 gels-11-00185-f005:**
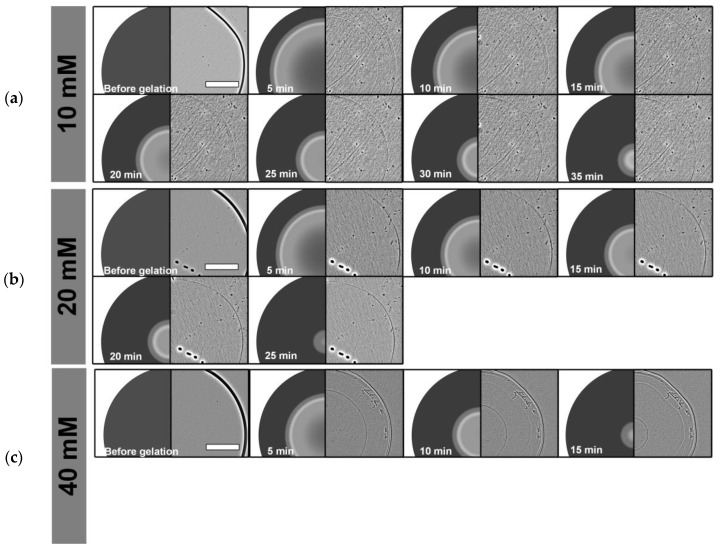
Time-lapse of alginate gelation simulation with different concentration boundary conditions of the cross-linking agent. The left half of each time point shows the visualization of the numerical model, while the right half shows the microscopic image of one experimental replicate. (**a**) 10 mM BaCl_2_ solution, (**b**) 20 mM BaCl_2_ solution and (**c**) 40 mM BaCl_2_ solution. Brighter areas indicate a higher amount of the ongoing gelling reaction. Scale bar indicates 1000 µm.

**Figure 6 gels-11-00185-f006:**
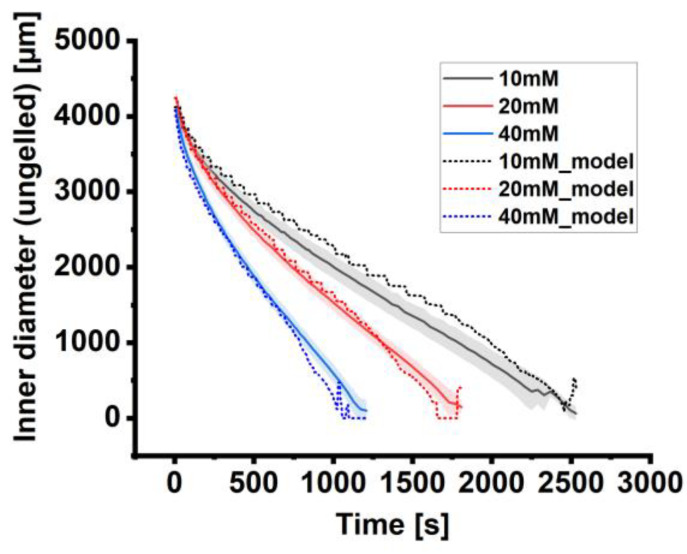
Comparison of experimental data (solid line) and numerical modeling (dotted lines).

**Figure 7 gels-11-00185-f007:**
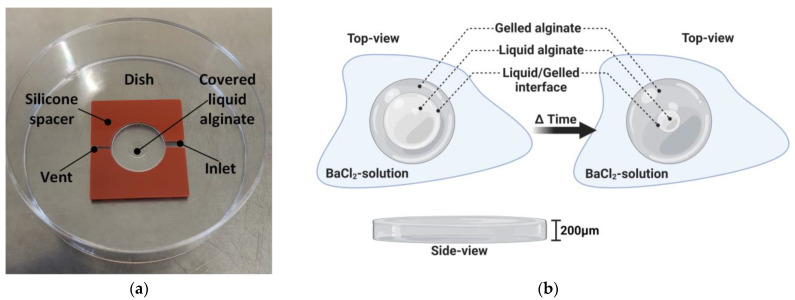
Setup and principle of observing the gelation process. (**a**) A thin disc-like volume of alginate is poured into a dish and covered by a thin silicone spacer for gelation with different BaCl_2_ solutions. This process can be observed using phase contrast microscopy, and a concentric decrease in the traveling liquid/gelled interface can be tracked and used for the quantification of the gelation process. (**b**) Schematic drawing at two different time points of alginate gelation. The disc-like volume of alginate is surrounded by the BaCl_2_ cross-linker solutions and, consequently, barium (and chloride) ions diffuse into the alginate sol, triggering the gelation that can be tracked by the traveling liquid/gelled interface over time. The diameters of the circular interfaces decrease over time and disappear after the complete gelation of the alginate discs. (**b**) generated with BioRender.com.

**Figure 8 gels-11-00185-f008:**
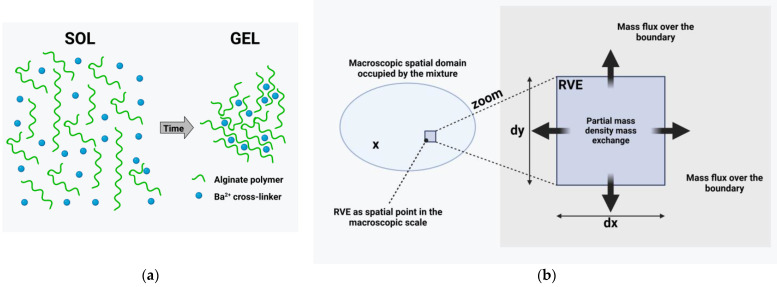
(**a**) Representative volume filled with the free polymer, barium ions and cross-linked polymer (and water). (**b**) Macroscopic domain and RVE as a magnification of a spatial point. The mass of constituent φ^α^ inside the RVE changes due to the flux over the boundary and the mass exchange. Created with BioRender.com.

**Table 1 gels-11-00185-t001:** Model parameters.

Variable	Parameter
$D_{0}^{poly}$	6.78 × 10^−12^ m^2^/s
$D_{0}^{ba}$	6.78 × 10^−9^ m^2^/s
$K^{poly}$	2.5
$K^{ba}$	2.3
$C^{poly}$	1.6 × 10^−1^
$C^{ba}$	8.0 × 10^−2^
$C_{\kappa }^{ba}$	2.5 × 10^−2^
$C^{\kappa }$	2.5 × 10^−2^

## Data Availability

The original contributions presented in the study are included in the article/Appendix A, further inquiries can be directed to the corresponding author.

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
