# Peer review of "Towards a Comprehensive Framework for Made-to-Measure Alginate Scaffolds for Tissue Engineering Using Numerical Simulation"

_gels, 2025, doi:10.3390/gels11030185_

Round 1
Reviewer 1 Report
Comments and Suggestions for Authors
The kinetics of the crosslinking by M2+ ions is a crucial subject in the characterisation of alginate hydrogels, and it is covered in the publication "Towards a comprehensive framework for made-to-measure alginate scaffolds for tissue engineering using numerical simulation." Overall, the paper provides a set of mixture theory-based kinetic equations, the derivation of which is presented in great detail. Additionally, the experimental results and the numerical solution of the equations correspond appropriately.
Before being published, a few things need to be clarified and improved. The research, for example, does not demonstrate how the numerical solution would yield data appropriate for material scientists and cell biologists planning to create scaffolds for various tissues. Actually, obtaining a crosslinked hydrogel with an elastic modulus that is similar to the target tissue's is one of the primary challenges. The modulus of the Ba-alginate system cannot be calculated using the numerical simulation.
Additionally, the authors asserted that the methodology outlined in their study can examine how debris (protein agglomerates) or small cargo (cells or 620 organoids) affect gelation behaviour. However, they point out in the discussion that more research should be done on various alginate compositions, cross-linking agent concentrations, and shrinkage.
In summary, the manuscript has potential, but several issues need to be addressed before it can be considered for publication. I recommend a major revision.
Author Response
We would like to thank Reviewer 1 for the valuable comments and suggestions and have answered everything in the attached pdf file.

Reviewer 2 Report
Comments and Suggestions for Authors
Alginate hydrogels are widely used in various biotechnological and biomedical applications. In particular, they are used as an immobilization matrix for enzymes or cells, carriers for the controlled release of pharmaceutical active compounds, wound dressings and dental impression materials. Alginates have unique physicochemical properties: they are non-toxic, biocompatible and gel under very mild conditions (ambient temperature, physiological pH, low-concentration and non-toxic cross-linking compounds). Alginates are based on linear alginic acid molecules, the molecular weight of which can vary over a very wide range. In recent decades, ultra-high viscosity (UHV) alginates have also been discovered, which have been recognized as promising biomaterials in regenerative medicine and tissue engineering. Unlike other commercially available alginates, UHV alginates have ten times longer polymers with the same cross-linking mechanisms and increased viscosity of the resulting hydrogels. The reviewed work studies the process of gelation of UHV alginates and the possibilities of its modeling based on experimental data. Given the poor study of gelation of UHV alginates along with the prospects for their use, the authors' research is relevant. However, the work in its original form is not ready for publication.
1. The disadvantage of the work is a very long introduction, which consists of 5 subsections and takes up 5 pages. The introduction should be shortened. It should contain the background of the studies considered below (in fact, this is point 1.1) and end with a clearly formulated objective of the work.
2. In the review material describing the main works on alginate gelation, on which the work is based, the cross-linking agents must be indicated. In addition, this material can be partially shortened and should be allocated to a separate section.
3. Alginate gelation is highly dependent on the cross-linking cations used. Typically, calcium, strontium or barium are used for gelation studies. Their cross-linking mechanisms are different. During the cross-linking process, different “egg box” cells are filled (https://doi.org/10.3390/ijms242216201). This fact should be taken into account and indicated.
4. It is not stated what motivated the choice of toxic barium chloride as a cross-linking agent. Commonly, calcium chloride or strontium chloride are applied for biomedical applications.
5. The alginate used is not characterized. The molecular weight of its components and the ratio of their M/G blocks are not specified. What was the original form of the alginate? Was it a powder or a solution? If a solution, the concentration must be specified. What is 1:1 mixture (vol%/vol%)? Why is the dynamic viscosity indicated in kPa∙s on the website Alginatec (https://www.alginatec.com/en/about-us/), while in the peer-reviewed work it is 3 orders of magnitude less?
6. The works underlying the Basic assumptions and further reasoning should be indicated. On line 345 there is the text “Following Truesdell’s metaphysical principles”, but there is no reference. On line 367 the text “cf. e.g. Truesdell [54], Bowen [55], etc.” appears, but references [54], [55] do not contain the indicated names. They are not visible in other references either.
7. What is new in equations (27)-(30)? Why were the parameters specified in Table 1 used for the calculations? What was the reason for their choice?
8. The conclusion should be rewritten. The conclusion lacks clarity in the description of the problems solved.
9. The list of references contains almost no publications from the last 5 years (7 out of 60).
Author Response
We would like to thank Reviewer 2 for the valuable comments and suggestions and have answered everything in the attached pdf file.

Reviewer 3 Report
Comments and Suggestions for Authors
This work presents a fundamental numerical simulation to guide alginate hydrogels preparation to support varying applications. The author evaluated the gelation speed as a function of the crosslinker concentration and the collected results fitted well to their numerical model. This work filled the gap of the current field and can help scientists to strategically design their hydrogel formulation for their research. This work can be published after addressing the following minor issues
The introduction is too long. It is suggested that the author should make it more concise.
Line 282, it is suggested that the author should discuss the rationale behind the chosen of barium chloride concentration. Why were 10, 20, and 40mM chose for the evaluation? The minimal and maximum value were chosen randomly because it didn’t matter or it is chose based on something? What is the stoichiometric ratio of the crosslinker vs the polymer under the above conditions?
Author Response
We would like to thank Reviewer 3 for the valuable comments and suggestions and have answered everything in the attached pdf file.

Round 2
Reviewer 1 Report
Comments and Suggestions for Authors
I appreciate the effort you have made in revising the manuscript and addressing the feedback provided. However, I must express my concern regarding the justification you provided about the objectives of the study.
I encourage you to either refine the objectives of the manuscript to align with the scope of the current work or provide a more comprehensive discussion that clearly demonstrates how the presented results contribute to addressing these objectives.
Author Response
Changes for Revision 1
“Towards a comprehensive framework for made-to-measure alginate scaffolds for tissue engineering using numerical simulation”
Alexander Bäumchen, Johnn Majd Balsters, Beate-Sophie Nenninger, Stefan Diebels, Heiko Zimmermann, Michael Roland and Michael M. Gepp
Reviewer 1
Q1.1: I appreciate the effort you have made in revising the manuscript and addressing the feedback provided. However, I must express my concern regarding the justification you provided about the objectives of the study.
We would like to thank reviewer 1 for giving very valuable comments. We think we could resolve the last concerns in this revision of the manuscript.
Q1.2: I encourage you to either refine the objectives of the manuscript to align with the scope of the current work or provide a more comprehensive discussion that clearly demonstrates how the presented results contribute to addressing these objectives.
We revised the section 1.5 (“Rationale predicting alginate hydrogel gelation“) in the introduction to refine the objectives of the current work. Parts, related to mechanical properties of the hydrogel (Young’s modulus) or software (e.g., “mobile app”) are moved to the section “Conclusion and outlook” since it will be part of futures studies and development of our proposed framework. With these changes the scope/rationale of the current work has been made more precise and improved the general structure of the manuscript.
Reviewer 2 Report
Comments and Suggestions for Authors
The authors have taken into account the comments made. This version of the manuscript can be published.

Author Response
Reviewer 2
Q2.1: The authors have taken into account the comments made. This version of the manuscript can be published.
We would like to thank reviewer 2 for giving very constructive comments, which were very helpful to significantly improve the quality of the manuscript.
Round 3
Reviewer 1 Report
Comments and Suggestions for Authors
The revisions made to Section 1.5 have successfully provided a more precise framework for the study, fostering a coherent connection between the introduction and the conclusions.
The manuscript in its current form makes a valuable contribution to the field and is deemed suitable for publication.